# The Development of 3D Primary Co-Culture Models of the Human Airway

**DOI:** 10.3390/ijms26115027

**Published:** 2025-05-23

**Authors:** Cinta Iriondo, Sem Koornneef, Kari-Pekka Skarp, Marjon Buscop-van Kempen, Anne Boerema-de Munck, Robbert J. Rottier

**Affiliations:** 1Department of Pediatric Surgery, Sophia Children’s Hospital, Erasmus Medical Center, 3015 CN Rotterdam, The Netherlands; m.iriondomartinez@erasmusmc.nl (C.I.); s.koornneef@erasmusmc.nl (S.K.);; 2Department of Cell Biology, Erasmus Medical Center, 3015 CN Rotterdam, The Netherlands

**Keywords:** complex in vitro cultures, lung, primary cells, in vitro model, human

## Abstract

Current animal and in vitro cell culture models do not fully recapitulate the physiological and pathophysiological characteristics of the human lung. As a result, the translation of these models to clinical practice is very limited, and clinical trials initiated on the extrapolation of such data fail. Although current models are beneficial in fundamental research, there is a need to constantly improve models to more accurately predict outcomes in clinical trials and personalized medicine. Here, we report important strategies to develop a 3D lung model with human primary lung cells. Starting from the well-established air-liquid interface (ALI) culture system, we describe a gradual increase in the complexity of the system by co-culturing different primary cell types, by testing different coatings, and by adding a three-dimensional matrix. As a result, we have established a reproducible 3D in vitro model of the airway consisting of human primary cells representing a differentiated mucociliary airway epithelium, an underlying submucosa with fibroblasts, and an endothelial interface.

## 1. Introduction

Respiratory diseases are one of the leading causes of morbidity and mortality globally, imposing a great health burden. According to the World Health Organization (WHO), respiratory diseases accounted for 3 out of the top 10 causes of death in 2019. In addition, infants and children are highly susceptible and vulnerable, especially to acute respiratory infections [1]. Respiratory diseases are poorly understood in children due to the lack of proper lung models. Although current models have made substantial contributions to the field, it is essential to develop in vitro models that more accurately reflect the complexities of the human lung microenvironment, such as involving multiple cell types and creating a three-dimensional structure, in order to gain deeper insights into the origin and subsequent progression of respiratory illnesses.

The cellular composition of the lung epithelium changes along the airway depending on the location and functionality. The proximal airways, which consist of the nasal cavity, trachea, bronchi, and bronchioles, warm up and humidify inhaled air and maintain the airways clean of pathogens and pollutants [2]. Basal cells are the airway cell progenitors and mainly differentiate towards secretory cells, which produce mucus to trap microbes and pathogens, and ciliated cells, which clear the airways [2,3]. The distal airways are formed by distal bronchioles and alveoli, which facilitate the exchange of oxygen and carbon dioxide. Alveolar type I cells are flat epithelial cells that cover 95% of the surface of the alveoli for optimal gas exchange, and alveolar type II cells are progenitor-like cells and produce surfactant to reduce surface tension, which allows lungs to re-expand during respiration [2,3].

The subepithelial layer consists of a multitude of cell types with different functions. Mesenchymal cells, such as fibroblasts, are important for the lung structure by producing extracellular components of the stroma. Smooth muscle cells surrounding the airways are important in mediating the contraction of the airways. Within the stroma, endothelial cells, perivascular cells, and smooth muscle cells form the blood vessels that transport oxygen and nutrients. Immune cells are also present, especially in the alveoli or in the airways during an acute infection, to neutralize pathogens [2]. In addition, the composition of the various cell types and of the extracellular cell–matrix may vary and can change under changing physiological conditions, such as diseases [4,5]. Due to this high complexity, improved models are needed that better recapitulate the physiology and pathophysiology of the human lungs.

Animal and 2D cell culture models are frequently used as models in basic and preclinical research [6]. Even though animal models offer natural 3D tissue structure, hemodynamics, and biomechanics, extrapolation to human physiology and disease is incomplete at best due to species-to-species differences. Simultaneously, clinical trials based on these preclinical models have a high failure rate [6,7]. Therefore, there is a need to develop models to better mimic human disease and with improved translatability.

Currently, one of the most common human in vitro airway models is the air-liquid interface (ALI) cultures, which consist of culture inserts that have a stiff, porous membrane made of polyethylene terephthalate (PET), polytetrafluoroethylene (PTFE), or polycarbonate (PC). These membranes are coated with extracellular matrix components, such as collagen, fibronectin, or laminin, to enhance the attachment of lung epithelial cells. These culture inserts can be exposed to air, which induces the epithelial cells to differentiate into a mucociliary epithelium [8,9]. The epithelial differentiation resembles the human airways, but these models lack the 3D structural complexity and cellular and interstitial physiology of the in vivo lung.

The biomedical and bioengineering fields are moving towards developing human organ models that are three-dimensional and involve different cell types to better mimic the in vivo conditions, with the primary objective to improve the ability to predict results in clinical trials and personalized medicine [6,10]. In the case of the proximal airways of the lung, models containing primary epithelial, mesenchymal, and endothelial cell (ECs) types are being developed to better study respiratory diseases and development, especially in pediatric lung diseases, due to the lack of proper models.

Here, we present data showing a step-wise increase in the complexity of the conventional ALI system to generate a primary three-dimensional human airway model. First, we optimized culture conditions to co-culture different cell types, and secondly, we selected cell culture hydrogels, suitable for our cells, to form a three-dimensional system.

## 2. Results

### 2.1. Selecting Co-Culture Medium for Primary Co-Culture of Lung Cells

In order to isolate and co-culture different primary cell types, it is instrumental to determine the optimal culture medium that supports the individual cell types in these complex cultures. Human primary bronchial epithelial cells (hPBECs), for instance, require serum-free medium because serum inhibits their proliferation and differentiation. Endothelial and mesenchymal cells, however, thrive in serum-rich medium (Table 1, Appendix A).

Therefore, various cell culture media and combinations thereof were tested for their capacity to support the different lung cell types (Figure 1 and Appendix A). We used KSFM and PneumaCultEX (StemCell Technologies Germany, Cologne, Germany) as references for evaluating the cultures grown in other types of media. An equal number of hPBECs were seeded, and cell proliferation was monitored daily by visual inspection. The hPBECs proliferated best in standard hPBECs media (KSFM or PneumaCultEX), less in hPBECs differentiation medium (BEGM or PneumaCultALI (StemCell)), and poorly in medium rich in fetal bovine serum (FBS, 10%), commonly used for ECs and fibroblasts. Naturally, human lung endothelial colony-forming cells (hECFCs) grew well in growth factor-enriched hECFC-medium (ECFC-EGM), and less in regular DMEM medium supplemented with FBS. Finally, human lung fibroblasts (hLFs) grew well in medium supplemented with FBS irrespective of additional growth factors present, such as ECFC-EGM or basic DMEM. Interestingly, hLFs also grew well in the different media for hPBECs, but ECs only survived for less than a week in these culture media, probably due to the lack of FBS and specific nutrients.

Next, different combinations of culture media were tested to evaluate the optimal conditions for a hPBECs-EC-hLF tri-culture. Of these three primary cell types, the hLFs are the least demanding cell type, as they appeared to grow equally well in fibroblast- or endothelial-specific medium. The complex tri-cultures should also be grown at ALI; hence, we first tested whether hPBECs grown submerged and at ALI could be maintained in a mixture of epithelial and endothelial medium.

Therefore, hPBECs were cultured in a 1:1 ratio of an epithelial-specific medium and the endothelial-specific ECFC medium to evaluate the proliferation and differentiation to a mucociliary epithelial layer in comparison to standard culture conditions (Figure 2). HPBECs grew poorly to a confluent cell layer when submerged in co-culture medium, suggesting that the 2D expansion of hPBECs should be performed in standard hPBEC growth medium, either KSFM or PneumaCultEX (Figure 2A). When hPBECs were subsequently grown at ALI in co-culture medium, they presented the characteristic cobblestone morphology of epithelial cells (Figure 2B). Since serum negatively affects the growth and differentiation of hPBECs, we further reduced the concentration of serum by mixing epithelial and endothelial medium at a 2:1 ratio.

HPBECs were grown in KSFM and subsequently in ALI in the 2:1 co-culture medium. Since hECFCs (grown in ECFC-EGM medium) and human lung microvascular endothelial cells (hMVECs) (grown in EGM-2MV medium) have slightly different medium requirements, we tested the co-culture medium of epithelium medium (BEGM) with either of the endothelium-specific media, i.e., BEGM:EGM-2MV or BEGM:ECFC-EGM in a 2:1 ratio. This resulted in maintenance and complete differentiation of hPBECs into a mucociliary epithelium as shown by the formation of tight junctions (ZO-1: Figure 3A, Figure 4B and Appendix A), ciliated cells (TUBIV: Figure 3A, Figure 4B, Appendix A) and goblet cells (MUC5AC: Figure 3B, Figure 4B and Appendix A).

Interestingly, we observed more ciliated (TUBIV: Figure 3A, Figure 4B and Appendix A) and goblet (mucus secretion, Figure 2B and MUC5AC: Figure 3B and Figure 4B) cells, and thicker epithelium layers (XZ images: Figure 3A,B) in co-culture medium, especially in the BEGM:EGM-2MV medium, than in standard medium, probably due to the cocktail of growth factors and hormones found in serum-rich co-culture medium.

Variations in differentiation were also observed among co-culture media that used different EC media (Figure 3A,B, and Appendix A). The ECFC-EGM medium and human umbilical vein endothelial cells (HUVEC)-specific medium (HUVEC-EGM, Appendix A) were almost the same, but ECFC-EGM lacks EGF, which is known to regulate hPBECs growth and differentiation when concentrations are between 0.5 and 10 ng/mL [11]. HMVEC-specific medium (EGM-2MV) also contains EGF, potentially explaining why hPBECs exhibited more differentiation in co-culture media containing either hMVEC- or HUVEC-specific medium. Another observation is the donor-dependent variation, a phenomenon very well documented in the literature (Figure 3A,B, Appendix A) [12]. This donor variability depends on multiple factors, such as age, gender, smoking history, obesity, or disease, which are difficult to disclose due to patients’ data privacy.

Having established a co-culture medium that supports the differentiation of hPBECS, we next focused on ECs and hLFs. HECFCs and hMVECs cultured on cell culture inserts in either EC-specific or co-culture medium for 14 days maintained the expression of endothelial markers CD31 (Figure 3C) and mesenchymal marker Vimentin (VIM) (Figure 3C,D). The presence of myofibroblasts, a cell type implicated in epithelial- and endothelial-to-mesenchymal transition (EMT and EndMT, respectively), was also examined by analyzing the expression of both alpha smooth muscle actin (ACTA2) and VIM. HECFCs of one specific donor had a small percentage of ACTA2 in both the endothelial and co-culture medium (Figure 3D). In addition, ECs also form tight junctions and express zonula occludens proteins that help maintain their cell layer integrity and regulate permeability. Therefore, transepithelial electrical resistance (TEER) values were also measured for hECFC and hMVECs in co-culture media (Figure 3E). No differences were observed between the endothelial and co-culture medium. Interestingly, hECFC expressed higher TEER values than commercial human lung microvascular endothelial cells (hMVEC-L, Lonza, Basel, Switzerland). The isolated and cultured hLFs also expressed the mesenchymal marker VIM, but did not express myofibroblast marker ACTA2 after 22 days in co-culture medium, indicating that they maintained their fibroblast phenotype (Figure 3F).

As the co-culture medium successfully supported the survival of the three different primary cell types intended to be used in a complex tri-culture (Figure 3), we next initiated co-cultures of hPBECs and ECs (Figure 4). First, the TEER was measured throughout the culture period to evaluate the integrity of the cell layer [13]. HPBECs showed TEER values that ranged from 300 to 1000 Ω × cm^2^ regardless of the condition (Figure 4A and Appendix A). Interestingly, hPBECs showed a high resistance early on at ALI when cultured in a co-culture medium, which rapidly reduced, but this was not observed when cultured in BEGM alone or co-cultured with hECFCs. This was not observed when hPBECs were co-cultured with HUVECs (Figure 4A and Appendix A). HPBECs also differentiated towards a mucociliary epithelium when co-cultured with either hECFCs or hMVECs at the basolateral side of the cell culture insert membranes (Figure 4B). Interestingly, hPBECs showed reduced differentiation towards ciliated cells in co-cultures rather than in mono-cultures (Figure 4B and Appendix A). Finally, both types of ECs remained viable during the cultures as deduced from the expression of CD31 and VIM after 14 days of co-culture (Figure 4C). In summary, the epithelial/endothelial co-culture medium (2:1) is most suitable for culturing primary hPBECs, hLF, and EC in a complex tri-culture (Table 1).

### 2.2. Selecting Surface Coating for Primary hPBECs-ECs Co-Cultures

Unlike cell lines, which mostly grow on uncoated cell culture plastics in basic medium supplemented with serum, primary cells have to grow in specialized media conditions, as previously discussed, and often also require specific surface coating based on mixing different ECM components to better adhere to the cell culture plastics.

The most commonly used surface coating to culture primary hPBECs is a mixture of bovine skin type I collagen (PureCol), human fibronectin, and bovine serum albumin (BSA), while a rat tail type I collagen-based surface coating is commonly used for primary ECs (Appendix A) [14,15]. However, we questioned whether this type of coating would also work for co-cultures on insert membranes since these are porous and generally made of other polymers, such as PET, PC, or PFTE.

Therefore, different coatings for hPBECs and ECs were tested, first on cell culture plates and then on PET insert membranes. Laminins were added to the coatings as they are one of the main components of the basement membrane. Laminin 521 (LN521), present in the lung, and laminin 332 (LN332), an epithelial-specific lamina, were used either alone or mixed with the hPBECs coating (Appendix A) to test if hPBECs and hECFCs would be maintained and proliferate on cell culture plates. The laminin coatings either showed the same or reduced support of cell proliferation, although laminin LN521 better supported cell adhesion than laminin LN332 (Appendix A). Interestingly, hECFCs cultured on hPBECs-specific surface coating on either plastic (Appendix A) or PET insert membranes (Appendix A) showed no differences in survival or growth compared to the control coating.

Based on these observations, the hPBECs-specific coating was selected to perform co-cultures on insert membranes, and different concentrations of LN521 were added to check whether there was an improvement in cell morphology, proliferation, and differentiation (Appendix A). Co-cultures of hPBECs and hECFC did not show obvious differences between the coating with and without LN521 in terms of hECFC cell morphology (Appendix A), and hPBECs differentiation towards mucociliary cells (Appendix A) after 14 days at ALI. Hence, we decided to only use surface coatings without laminins.

Because no variations in hECFC cell cultures on either hPBECs (Appendix A) or EC coating (Appendix A) were observed, the hPBECs coating was tested to culture hMVECs on inserts in co-culture with hPBECs (Figure 5A,B). HMVECs in hPBECs coating appeared similar in terms of cell size, morphology, and attachment compared to hMVECs grown on EC coating (Figure 5A). The type of coating used to culture the hMVECs had no effect on the differentiation of hPBECs to a mucociliary epithelium either (Figure 5B), so the hPBEC-specific coating was selected for future co-cultures.

Despite hMVECs forming an intact cell layer during extended culturing periods, maintaining the integrity of a monolayer, as evidenced by CD31 positivity, was inconsistent over time. To improve the maintenance of the attachment of ECs to porous PET membranes, the inserts were coated overnight (ON) and air-dried (see materials and methods). Moreover, the addition of medium flow was tested (Figure 3C). HMVECs were plated on the basolateral side of the membranes of inserts, and the inserts were placed in the Simple-Flow device [16]. The Simple-Flow device contains medium flow at the basolateral side and thus allows hMVECs to be in direct contact with the shear stress forces induced by the flow. As shear stress also affects hMVECs attachment to the cell culture surfaces, this would show which coating conditions worked best for hMVECs attachment.

HMVECs appeared to attach better if the time of coating the cell surface was extended from 2 h to overnight, and if the culture conditions were static rather than fluidic. In addition, once medium flow was added, the air-drying step had a positive impact on hMVECs cell attachment, and fewer CD31 connections and cells were lost (Figure 3C).

In summary, a type I collagen-based coating was most optimal for the hPBECs-hMVECs co-cultures, and the insert membranes were coated for an extended time and then allowed to air-dry.

### 2.3. Selecting a Three-Dimensional Hydrogel for In Vitro Cultures

In order to further mimic the physiological structure of the human airway, which also contains a subepithelial matrix with supportive cells, a three-dimensional matrix containing hLFs was tested to form an interstitial space between hPBECs and ECs [17].

A requirement of this interstitial tissue layer is maintaining structural integrity for culture periods up to 2 weeks, so hPBECs could fully differentiate into mucociliary epithelium on top of this 3D matrix. Therefore, several synthetic scaffolds were explored as they generally have higher mechanical strength than natural hydrogels (Appendix A) [18].

First, Noviogel-P5K and Noviogel-RGD5K were tested, consisting of a thermosensitive synthetic polymer called polyisocyanopeptide (PIC) [19], while the RGD5K gel variant contains the RGD peptide motif to facilitate integrin-dependent cell adhesion. Different concentrations of Noviogel-P5K and Noviogel-RGD5K were prepared, and hPBECs were cultured on top of them. However, instead of forming a monolayer, the hPBECs formed aggregates (Appendix A). Initially, Noviogel-RGD5K showed a slightly better attachment than Noviogel-P5K, probably due to the RGD peptide, but gradually most of the hPBECs detached. The remaining hPBECs were viable, even after 17 days in culture, as shown by Calcein AM staining (Appendix A). Next, hLFs were incorporated into the Noviogel, but the cells did not form the classical spindle-shaped fibroblast appearance, indicating either that hLFs attached poorly to the gel or the gel was too soft (Appendix A).

Next, natural matrix type I collagen gel (PureCol) was added to the Noviogel to increase potential support for hLFs in the hPBECs-hLFs co-cultures. In addition, other ECM components, such as human fibronectin and LN521, were also mixed with the Noviogel to test whether they further supported cell attachment (Appendix A). However, none of these added components appeared to have a positive effect on the hLFs or the hPBECs, except for the addition of collagen. HPBECs and hLFs attachment improved further when only type I collagen hydrogels were used.

Other synthetic polymers, such as FN-4RepCT (FN-Silk, Spiber Technologies, Stockholm, Sweden, Appendix A), Manchester Biogels, Alderley Edge, United Kingdom (Appendix A), and GrowDex (UPM Biomedicals, Helsinki, Finland, Appendix A), were tested, but all showed poor attachment of hPBECs and hLFs on top of these gels and poor visibility of the cultured cells. FN-Silk foam technology, based on spider web silk proteins [20], formed many configurations, but it was challenging to reproducibly produce a straight surface to culture cells onto (Appendix A). However, unlike the hPBECs, hLFs would attach to the FN-silk (Appendix A).

Therefore, type I collagen gels were tested as the basis for the complex tri-cultures, because type I collagen is the most abundant ECM protein in the normal lung, and collagen-based hydrogels are the most widely used in respiratory tissue engineering [21]. HPBECs were plated either on top of collagen gels as mono-cultures (Appendix A) or on top of an hLF containing layer of collagen (Figure 6A,B and Appendix A). In both cases, hPBECs formed a completely differentiated mucociliary layer on top of this hLF-collagen matrix, and in the latter case, hLFs formed a typical elongated morphology in the matrix. After 14 days at ALI, the mucociliary epithelium had tight junctions (ZO-1), and contained basal cells, exemplified by the expression of TP63, Keratin 5 (KRT5) and Keratin 8 (KRT8), as well as fully differentiated ciliated (TUBIV) and goblet (MUC5AC and MUC5B) cells (Figure 6B).

Even though hPBECs’ adhesion improved considerably on the collagen-based gel supports, hLFs contracted collagen hydrogels frequently (Appendix A). The density, strength, and mechanical integrity of the collagen fibers were slightly increased by mechanically compressing the gels at deposition and by absorbing the excess buffer through capillary action. Although this extended the time of culturing for up to 14 days at ALI (Figure 6A), it still had disadvantages, as the gel surface was slightly ruffled, and fibers of the gauze used to reabsorb excess buffer were left on top of the gels (Appendix A). TEER values were measured during ALI, and, generally, when hPBECs were plated on collagen matrices, much lower TEER values were generated than when hPBECs were plated on standard insert membranes (Appendix A). This most likely reflects the weaker adhesion forces applied onto the soft gel, leading to reduced cell–cell contacts as well [22].

To overcome the contractility of the hLFs, a broad-spectrum metalloprotease (MMP) inhibitor, GM6001 (Ilomastat), was tested. Indeed, the addition of GM6001 prevented hLFs from contracting the collagen gels (Appendix A), without affecting the differentiation of hPBECs towards ciliated cells (Appendix A). Finally, combining the experimental data led to the initiation of tri-cultures with hPBECs, hLFs, and hECS (Figure 6C). After 14 days at ALI, the cultures clearly displayed a mucociliary epithelium on top of the collagen matrix, as evidenced by the presence of ciliated (TUBIV, FOXJ1; Figure 6D,H) and goblet cells (MUC5B, MUC5AC, Figure 6D,E,I). Within the collagen matrix, mesenchymal type III intermediate filament marker VIM-positive hLFs were present (Figure 6E,J), and ERG-1 and VIM-positive hMVECs were at the basal side of the membrane (Figure 6F,G,K).

## 3. Discussion

We aimed to develop a three-dimensional model of the human airway using primary human bronchial epithelial cells, lung ECs, and a 3D matrix containing hLFs to closely mimic the structural anatomy of the human airway. This would provide a model to obtain better insights into the pathophysiology of pediatric respiratory diseases and to more accurately predict outcomes.

Primary cells are known for preserving their genomic integrity and maintaining most of their phenotypic characteristics for a limited number of passages. However, primary cells also require specialized cell culture media with defined supplements to preserve their identity [23,24]. In order to co-culture different primary cell types together, we first identified the optimal medium to support the survival and growth of these primary cells in advanced cultures. We observed variations in epithelium thickness and hPBECs differentiation rates when using different EC-specific media in the co-culture medium. HECFC-specific medium lacked EGF, a component known to regulate hPBECs’ growth and differentiation, resulting in reduced numbers of ciliated cells compared to other EC-specific media containing higher amounts of EGF [25]. Interestingly, variability in hPBECs cell morphology and differentiation was also observed previously between using BEGM or PneumaCultALI [26], probably caused by variations in the concentrations of retinoic acid (RA) and EGF growth factors [11,27].

We showed that mixing epithelial- and endothelial culture media in a 2:1 ratio efficiently supported hPBECs-ECs-hLFs tri-cultures. A mixture of epithelial and endothelial-specific media in a 1:1 ratio for primary co-cultures has been commonly reported [15,28,29,30,31,32]. For instance, Sellgren et al. also tested different media to culture ECs in co-culture [32]. They showed that for their tri-cultures, a medium composition similar to ours, but for long-term co-cultures, the medium needed to be optimized to further improve the quality of co-cultures. We show that using a 2:1 ratio, which reduced the serum concentration, supported long-term cultures.

Since plasma-treated culture supports are not sufficient to support the attachment of primary cells, coatings mainly containing ECM components, such as collagen and fibronectin, are added to the surface of the vessels to improve cell adhesion. We used a mix of type I collagen, human fibronectin, and BSA to coat hPBECs surfaces, as reported [33]. Alternatives are rat tail type I collagen [34] or type IV collagen [15,33]. Type I collagen [32,35,36], fibronectin [37,38] and gelatin-based [39,40] coatings are frequently used to coat cell culture vessels for eECs, as well as type IV collagen [41].

Apart from the type of coating used, the method of application also influences its adherence to cell culture vessels. Dupont-Gillain and colleagues have shown that the amount of collagen absorbed by surfaces and its supramolecular organization vary depending on the surface material and treatment, collagen concentration, duration of absorption, and drying step conditions [42]. For instance, they showed that the organization of type I collagen absorbed on PS depended on the adsorption duration: the adsorbed amount of collagen on the surface increased over time. The adsorption duration also affected the structure of the collagen fibers by reorganizing them, leading to the formation of netlike structures [42,43]. They demonstrated that netlike structures were also formed when the collagen layer was dried at a slow rate on PS surfaces. These discontinuous collagen networks enhanced HUVECs’ spreading and cytoskeleton organization on the surface [43,44,45]. Others showed that collagen oxidizes when exposed to air, facilitating the crosslinking of collagen. As a consequence, this led to poor collagen fibril formation and increased substrate stiffness, causing the spreading of chondrocyte cells on substrates [46]. We showed that extending the time of the coating and drying the collagen-coated surface to air-dry likely resulted in improved hMVEC attachment to the insert membranes, most likely due to alterations in the amount of collagen absorbed by the insert membranes and structural changes in collagen fibrils.

Human lungs have a three-dimensional configuration and stiffness that differs from the cells cultured on 2D plastic surfaces. In 2D, cells display a flattened shape with abnormal distribution of ligand complexes, which is not observed in 3D structures [47]. In addition, the stiffness (measured as Young’s elastic modulus, *E*) of cell culture plastic or glass is 2–4 GPa, while acellular collagen hydrogels are about 0.5–1.2 kPa, which resembles more closely the *E* of normal lung tissue of 0.2 kPa [48]. Thus, in order to better recapitulate the lungs’ physiological conditions, 3D ECM matrices need to be taken into account in in vitro models. Collagen matrices, particularly at a concentration of 2 mg/mL, are among the most commonly used ECM materials in tissue engineering [49], and a similar range was utilized in this study with uncompressed collagen gels (2.4 mg/mL). Pageau et al. reported that a range of 2–3 mg/mL of collagen gel concentration was optimal for hPBECs-hLFs co-cultures and to prevent shrinkage produced by fibroblasts [50]. However, we observed that hLFs still contracted collagen gels at this range of collagen concentration, a phenomenon commonly observed and well known in the literature [51,52,53]. Increasing the collagen concentration and density, and thus stiffness, could have been one option to prevent shrinkage caused by hLFs. However, gels with high collagen concentration impair cell migration and viability [54]. In addition, an increased ECM density and stiffness are also hallmarks for disease and cancer [4,5,55]. Fibroblasts deposit the ECM and secrete ECM-modifying enzymes, such as MMPs, that catabolize different components of the matrix, such as collagen [56]. Thus, to prevent the contraction of the gels by the hLFs, a broad MMP inhibitor (Ilomastat) was added [57]. Ishikawa et al. used Ilomastat in co-cultures of hPBECs and the human fetal lung fibroblasts cell line IMR-90 to study the effects of TGF-β1, which induces the transdifferentiation of fibroblasts to myofibroblasts, becoming αSMA+ [58]. They maintained the co-cultures for up to 21 days without contraction of the collagen gel, and the Ilomastat did not affect hPBECs differentiation. Choe et al. also used Ilomastat to prevent IMR-90 from contracting collagen gels in their hPBECs-IMR-90 co-cultures, which lasted for up to a month, and hPBECs also differentiated towards a mucociliary epithelium [59,60].

HPBECs were seeded onto a collagen gel containing fibroblasts and were air-exposed at ALI, which led to the differentiation towards a mucociliary epithelium. Others also reported similar mucociliary differentiation on their 3D models of the human airway [58,61]. In general, our tri-cultures showed lower TEER values than co- or mono-cultures on day 14 at ALI, and slightly lower than previously reported [61]. Interestingly, a TEER peak was observed on day 0 at ALI every time co-culture medium was used, highlighting the importance of the type of co-culture medium used for our cultures and the effect it has on cells. In addition, we experienced differences in TEER values that were dependent on the donor and passage number, culture medium, and variables related to the technical details, such as the angle and position of the electrode [62].

Although we report significant progress and novelty to human in vitro lung cultures, there are still caveats. One such caveat is the lack of immune cells in the cultures, which would be a logical next step in adding complexity to the cultures. Moreover, as we indicated, another improvement would be to include dynamics to these cultures.

In summary, we described the establishment of a 3D in vitro model using primary cells derived from human resected lung material. It contains a fully differentiated mucociliary airway epithelium derived from primary epithelial cells, and it uniquely includes a submucosa-like layer with primary fibroblasts and an interface of primary endothelial cells. This human airway model of primary 3D tri-cultures more accurately mimics the human airway and may be used to study airway development, repair, and disease responses.

## 4. Materials and Methods


**Patient Samples**


Tumor-free human lung tissue and bronchus rings were obtained from non-tumorigenic lung resection material of patients diagnosed with lung cancer requiring surgery at the Erasmus University Medical Center (Rotterdam, The Netherlands) and approved by the Medical Ethical Committee (METC nr. MEC-2012-512). All lung samples collected for our research followed the “Human Tissue and Medical Research: Code of conduct for responsible use (https://www.coreon.org/wp-content/uploads/2020/04/coreon-code-of-conduct-english.pdf, accessed on 3 January 2024) guidelines. All samples were anonymized, and the age and sex of the donors are listed in Appendix A.


**Primary cell culture**


Human primary bronchial epithelial cells (hPBECs), human microvascular endothelial cells (hMVECs), human endothelial colony-forming cells (hECFCs), and human lung fibroblasts (hLFs) were isolated and cultured as described (manuscript submitted). Briefly, all cell types were extracted from tumor-free lung resection tissue. HPBECs were isolated from a human bronchus after treatment with 0.18% Protease type XIV (Sigma-Aldrich, Houten, The Netherlands, Cat. No P5147-1G) in DPBS at 37 °C for 2 h and seeded in coated 6-well plates using complete KSFM medium. HMVECs and hECFCs were isolated and cultured according to the Alphonse et al. protocol [35], but without the cloning cylinders for hMVECs. Both endothelial cell types were further expanded and MACS-purified until reaching a >97% CD31+ cell population. HECFCs were cultured in hECFC-specific medium (ECFC-EGM). HLFs were isolated by mincing very small pieces of lung tissue attached to 6-well plates in 1 mL DMEM + 10% FBS medium, and hLFs emerged within 2 weeks. Human Lung microvascular endothelial cells (hMVEC-L) and human umbilical vein endothelial cells (HUVECs) were both obtained from Lonza (Table 2). All details for cell culture media, coatings, and trypsinization solutions used for each cell type can be found in Appendix A.


**Mono-culture set up**


HPBECs were thawed and expanded to 95% confluency in coated 10 cm dishes using complete KSFM medium. Then, hPBECs were plated on coated 12 mm inserts (Costar, Corning, Arlington, VA, USA, Cat. No 3460; coating: see Appendix A) at a cell density of 1.5 × 10^5^ cells/cm^2^ and put at ALI once they were confluent using complete BEGM with 50 nM EC-23 (retinoid acid agonist). HPBECs were allowed to differentiate at ALI for 2 weeks, and the media were changed 3 times a week.


**Co-culture set up**


12 mm cell inserts (Costar, Corning, Arlington, VA, USA Cat. No. 3460) were coated with 1 mL co-culture coating (Appendix A) on the basolateral side of the membrane overnight (ON) at 37 °C. The next morning, we removed the coating, and the inserts were left to air-dry upside-down on top of the lid in a sterile cell culture hood for 2–3 h to improve endothelial cell attachment. HMVECs were seeded on the basolateral side of the insert at a cell density of 2 × 10^5^ cells/cm^2^ in 100 µL of complete EGM-2MV medium (Appendix A). Once the drop was added, the cell culture plate was placed on top of the upside-down inserts (placed on the lid), and the hMVECs were allowed to attach to the membrane at 37 °C for 3.5–4 h. Then, inserts were flipped back, and 1 mL of complete EGM-2MV medium was added to the basolateral compartment only for 4 days. Then, the apical side of the insert membrane was coated with the same coating at 37 °C for 2 h and hPBECs were seeded at a cell density of 1.5 × 10^5^ cells/cm^2^ in 400 µL complete BEGM (Appendix A) with 1 nM EC-23 (Tocris, Bristol, UK, Cat No. 4011, 10 mg) and 5 µM ROCK inhibitor (Y-27632, Abmole, Huissen, The Netherlands, Cat No. Y-27632). EGM-2MV medium was replaced with co-culture medium BEGM:EGM-2MV (2:1 ratio) in the basolateral compartment. Once hPBECs reached 100% confluency (after 3–4 days), hPBECs were put at ALI, and 1 mL BEGM:EGM-2MV (2:1 ratio) with 50 nM EC-23 co-culture medium was added to the basolateral compartment, which was refreshed 3 times a week.


**Tri-culture set up**


HMVECs were plated on the basolateral side of the membrane as described above. Collagen gel solution (Cellmatrix type I-A (Nitta Gelatin, Osaka, Japan): 10X MEM (Gibco, Bleiswijk, The Netherlands): Reconstitution Buffer (Nitta Gelatin), ratio 8:1:1) was prepared and put on ice. Then, hLFs were seeded at a density of 5 × 10^3^ cells in 200 µL of collagen gel solution at the apical side of the membrane and incubated for 30–60 min at 37 °C. Once solidified, 500 µL EGM-2MV medium was added to the apical compartment, and the hMVEC-hLFs were cultured for 3–4 days. Then, hPBECs were seeded on top of the gel as described, but using a cell density of 2.5 × 10^5^ cells/cm^2^ in 500 µL co-culture medium BEGM:EGM-2MV (2:1). EC-23 and RI were also used, and the basolateral compartment was also replaced with co-culture medium. Once hPBECs reached confluency, the co-cultures were put in ALI for 14 days. To prevent collagen gels from shrinking due to hLFs, collagen gels were flattened as explained (manuscript submitted), and/or 30 nM GM6001 (Tocris, Bristol, UK, Cat No. 2983) was added in the co-culture medium during ALI.


**Calcein-AM assay**


This assay was used according to the manufacturer’s instructions. Briefly, 50 µg lyophilized calceinAM (Red-Orange, Bleiswijk, The Netherlands) tubes (ThermoFischer, Bleiswijk, The Netherlands, Cat. No C34851) were used and diluted first in DMSO, and then in cell culture medium to a final concentration of 5 µM. Cells were placed back in the incubator, and after 10–15 min, pictures were taken using an epifluorescence microscope.


**Trans-epithelial electrical resistance (TEER)**


500 µL DPBS with Mg^2+^ and Ca^2+^ was added to the apical compartment of the inserts. TEER values were measured just as previously described [14,63]. The background value was subtracted from the TEER values measured. TEER measurements plots were created with GraphPad Prism 9.


**Paraffin embedding of tri-cultures**


Insert membranes containing the tri-cultures were washed once with DPBS plus Mg^2+^ and Ca^2+^ (Gibco, Cat. No. 14287-072) to prevent the disruption of tight junctions and were fixed with 4% (*w*/*v*) paraformaldehyde (PFA; Sigma-Aldrich, Houten, The Netherlands, Cat. No. 441244) at room temperature (RT) for 1 h and embedded in Histogel (ThermoFisher). Samples were subsequently paraffin-embedded and sectioned into 5 µm slices using the microtome. Sliced samples were placed onto slides (Menzel-Gläser, Ismaning, Germany SuperFrost^®^ Plus) containing 10% ethanol (EtOH) and placed on a heating plate at 40 °C to allow EtOH to evaporate.


**Hematoxylin and Eosin (H&E) staining**


Slides containing 5 µm sections of tri-culture were processed as shown in Appendix A. Then, samples were mounted with Pertex, and a coverslip (Menzel-Gläser 24 × 60 mm) was added on top. Samples were left to dry ON in the fume hood.


**Immunofluorescence of tri-cultures paraffin sections**


Sections were deparaffinized by 3 xylene washes of 3 min each, followed by rehydration in an ethanol series in distilled water. Meanwhile, TE buffer pH 9 (10 mM Tris (Sigma-Aldrich, Cat. No T6066-5KG), 1 mM EDTA (Sigma-Aldrich, Cat. No E1644-1KG) in Milli Q water) was prepared, and 500 mL of TE buffer was added to a glass beaker. Antigen retrieval was carried out by boiling the slides in TE buffer at 600 W for 15 min, and standard immunofluorescence staining was performed as described previously [14,63]. Samples were placed in a humidity chamber and incubated with primary antibodies (Appendix A), secondary antibodies (Appendix A), and 1:2000 DAPI solution (BD Biosciences, Cat. No. 564907). Finally, mounting media (2.4% (*w*/*v*) Mowiol (Sigma-Aldrich, Houten, The Netherlands, Cat. No. 81381), 12% (*w*/*v*) 0.2 M Tris pH 8.5 (Sigma-Aldrich, Houten, The Netherlands, Cat. No. T6066), 4,75% (*v*/*v*) glycerol (Honeywell, Delft, The Netherlands, Cat. No. 49770-1L) in ddH_2_O) was added, and slides were covered with coverslips (Menzel-Gläser, Ismaning, Germany 24 × 60 mm). Samples were imaged with the ECHO Revolve fluorescence microscope and Leica SP5 confocal microscope. Images were processed with Image J, version 1.54g.


**Immunofluorescence of insert membranes**


Insert membranes without gels were fixed with 4% PFA (*w*/*v*) at RT for 10 min and permeabilized with 0.1% (*v*/*v*) Triton-X100 (Sigma-Aldrich, Cat. No. T8787-25) in DPBS (0.1% T-PBS). Samples were blocked with 1% (*w*/*v*) BSA (Roche Diagnostics, Basel, Switzerland) and 5% (*v*/*v*) normal donkey serum (EMD Millipore, Darmstadt, Germany Cat. No. S30-100 ML) in 0.1% T-PBS at RT for 30 min and incubated with primary antibodies (Appendix A) in blocking buffer at 4 °C ON. Then, membranes were incubated with secondary antibodies (Appendix A) and 1:2000 DAPI solution in blocking buffer at RT in the dark for 2 h. Finally, membranes were put in DPBS, mounted on slides with mounting media, and covered with coverslips.


**Whole mount immunofluorescence staining of tri-cultures**


Insert membranes containing the tri-cultures were stained as described above, but volumes and times were changed. Differences can be found in Appendix A. Finally, the tri-cultures were mounted on slides according to the protocol of Dekkers et al., and fructose-glycerol clearing agent was also used [64]. The samples were directly stored at 4 °C and imaged with an ECHO Revolve fluorescence microscope and a Leica SP5 confocal microscope. Images were processed with Image J.

## Figures and Tables

**Figure 1 ijms-26-05027-f001:**
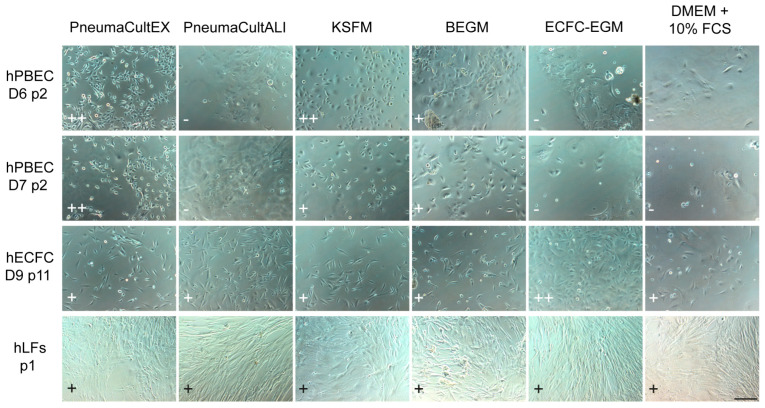
The effect of different media on cell morphology and proliferation. Bright field images of cultures of 2 independent hPBEC donors (D6, D7, p2), hECFCs (p11), and hLFs (p1) on day 3 of mono-culture in different hPBEC (PneumaCultEX, PneumaCultALI, KSFM, BEGM), endothelial (ECFC-EGM), and hLF (DMEM + 10% FBS) media. HPBECs were plated at a 1 × 10^4^ cells/cm^2^ density, and hECFCs and hLFs at 5 × 10^3^ cells/cm^2^ density. Scale bar = 80 µm. Scoring, indicated in the lower left corner, is defined as: ++ very good (high confluency, minimal cell patches, proper cell morphology), + good (medium confluency, some cell patches, decent cell morphology), and − bad (low confluency, many cell patches, bad cell morphology).

**Figure 2 ijms-26-05027-f002:**
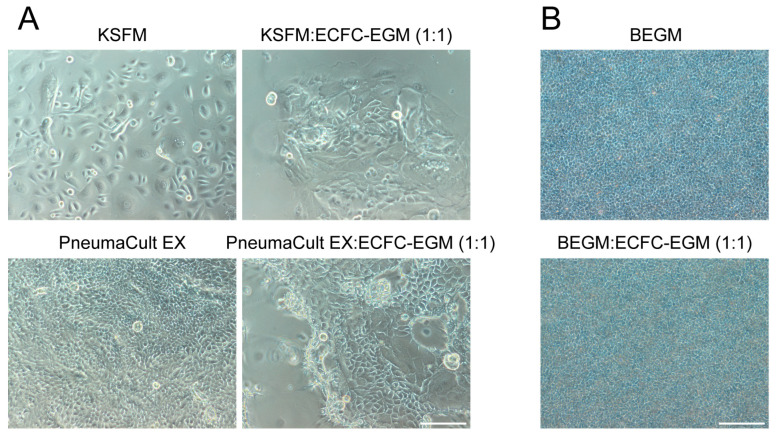
The expansion of hPBECs is better in epithelial-specific mediums. (**A**) Bright field images showing hPBECs (Donor 10, p3) in hPBEC growth media (KSFM or PneumaCultEX) or co-culture medium. Images were taken on day 4 of culture, and hPBECs were plated on cell culture plates at a density of 1 × 10^4^ cells/cm^2^, N = 2. (**B**) Bright field images of hPBECs (Donor 6, p2) after 14 days at ALI. HPBECs were grown on insert membranes until confluency with BEGM and then air-exposed in either BEGM or co-culture medium BEGM:ECFC-EGM (1:1). Scale bar = 80 µm, N = 3.

**Figure 3 ijms-26-05027-f003:**
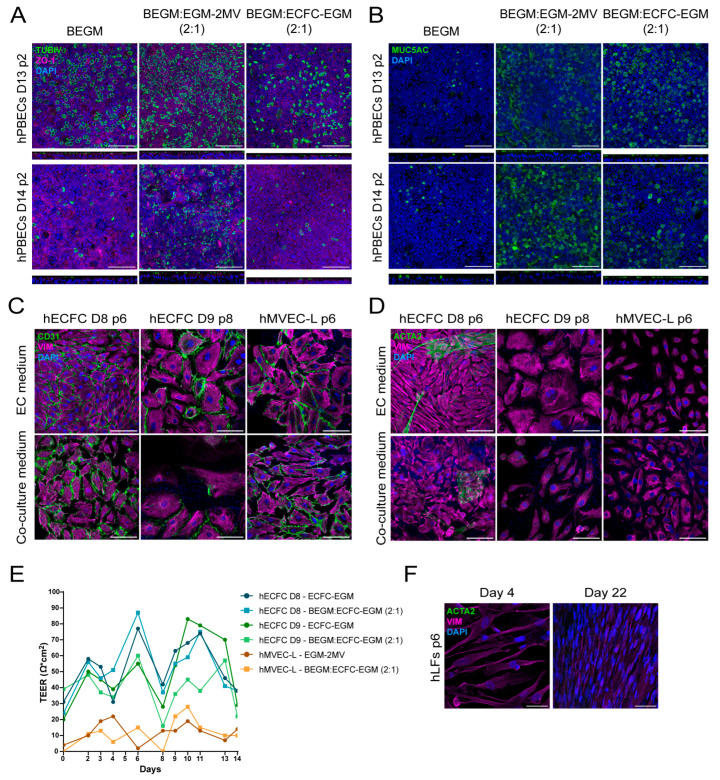
Testing co-culture media on mono-cultures of primary lung cells. (**A**,**B**) Confocal images showing differentiated hPBECs of two different donors (Donor 13 and 14, both p2). (**A**) ciliated cells (TUBIV, green), tight junctions (ZO-1, magenta) and nuclei (DAPI, blue), and (**B**) goblet cells (MUC5AC, green), and nuclei (DAPI, blue). XZ images are shown below the individual images to appreciate cell layer thickness. Scale bar = 100 µm, N= 3–5. (**C**,**D**) hECFC cells (Donor 8, p6; Donor 9, p8) or hMVEC-L (p6) in control or co-culture medium (BEGM:ECFC-EGM 2:1). (**C**) Endothelial cells expressed CD31 marker (green), VIM (magenta) and DAPI (blue), and (**D**) αSMA (green), VIM (magenta) and DAPI (blue) after 14 days on cell culture inserts. Scale bar = 100 µm, N = 3. (**E**) TEER values (Ω × cm^2^) for two hECFC donors (Donor 8 and 9) and hMVEC-L in either endothelial or co-culture medium. (**F**) Human lung fibroblasts in co-culture medium on day 4 and day 22 of culture stained against αSMA (green), VIM (magenta), and DAPI (blue). Scale bar = 50 µm.

**Figure 4 ijms-26-05027-f004:**
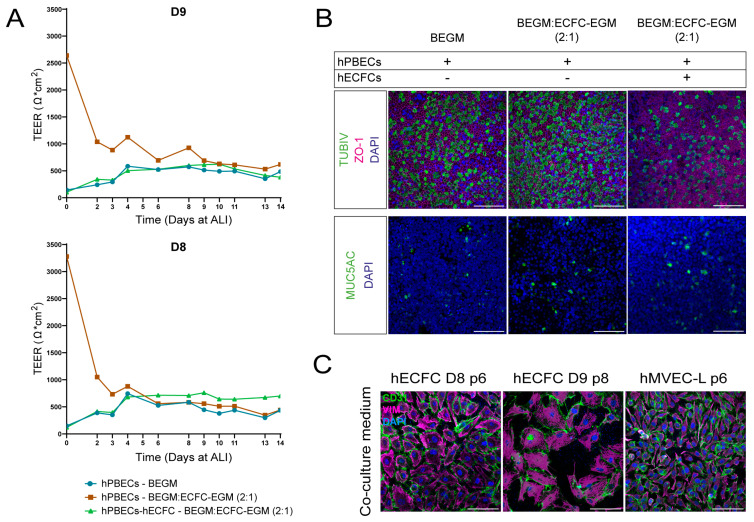
Testing co-culture medium for hPBECs-hMVECs co-cultures. (**A**) TEER values (Ω × cm^2^) of two hPBEC donors (Donor 8 and 9, both p2) mono-culture in BEGM (circle, blue line), hPBECs mono-culture in BEGM:ECFC-EGM (2:1) co-culture medium (square, brown line), and hPBECs-hECFC co-culture in co-culture medium (triangle, green line) measured on different days at ALI. (**B**) Differences in hPBECs differentiation between mono- and co-culture with hECFCs. BEGM and BEGM:ECFC-EGM (2:1) co-culture medium was used. Scale bar = 100 µm. (**C**) hECFC cells (Donor 8, p6; Donor 9, p8) or hMVEC-L (p6) as co-cultures with hPBECs in co-culture medium (BEGM:ECFC-EGM 2:1). Endothelial cells expressed CD31 marker (green), VIM (magenta), and DAPI (blue) after 14 days on cell culture inserts. Scale bar = 100 µm, N = 3.

**Figure 5 ijms-26-05027-f005:**
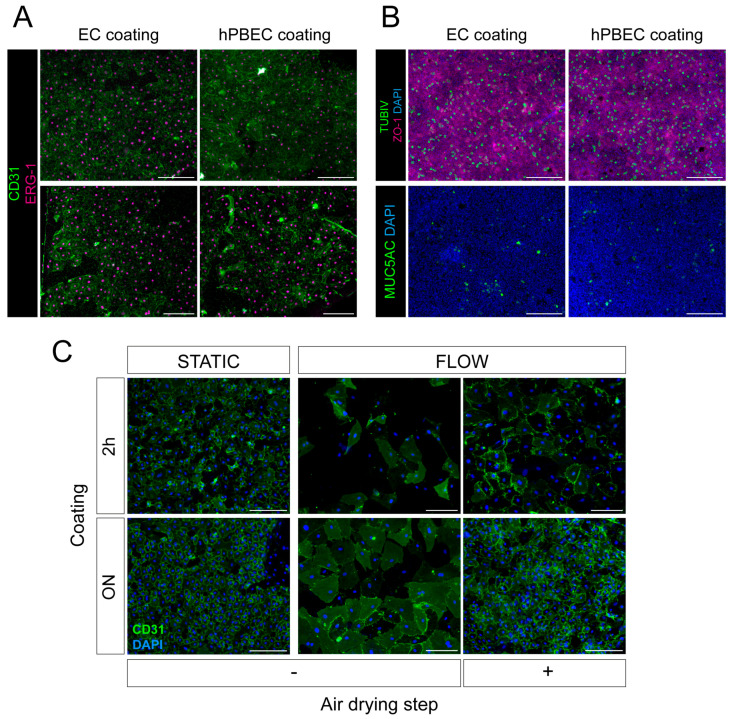
Testing coating for endothelial cells plated on insert membranes. (**A**) Images of hMVEC (Donor 2, p6) in co-culture with hPBECs Donor 2 p3 using BEGM:EGM-2MV (2:1) medium. HMVECs were stained for endothelial markers CD31 (green) and ERG-1 (magenta). HPBECs and endothelial cell (EC) coatings were used (2 images/condition). Scale bar = 200 µm, N = 3. (**B**) Images of hPBECs from Figure 3B co-cultures. HPEBCs were stained for ciliated (TUBIV) and goblet (MUC5AC) cells, as well as tight junctions (ZO-1). The apical side of the inserts (hPBECs) was coated with hPBEC coating, and the basolateral side (hMVECs) was coated either with hPBEC coating or EC coating. Scale bar = 200 µm, N = 3. (**C**) Images of hMVEC Donor 16, p9 in EGM-2MV medium stained against CD31 (green) and DAPI (blue). hPBECs/co-culture coating condition was used in static (0 µL/min) or flow (200–400 µL/min) conditions in combination with or without the air-drying step. Scale bar = 200 µm, N = 2.

**Figure 6 ijms-26-05027-f006:**
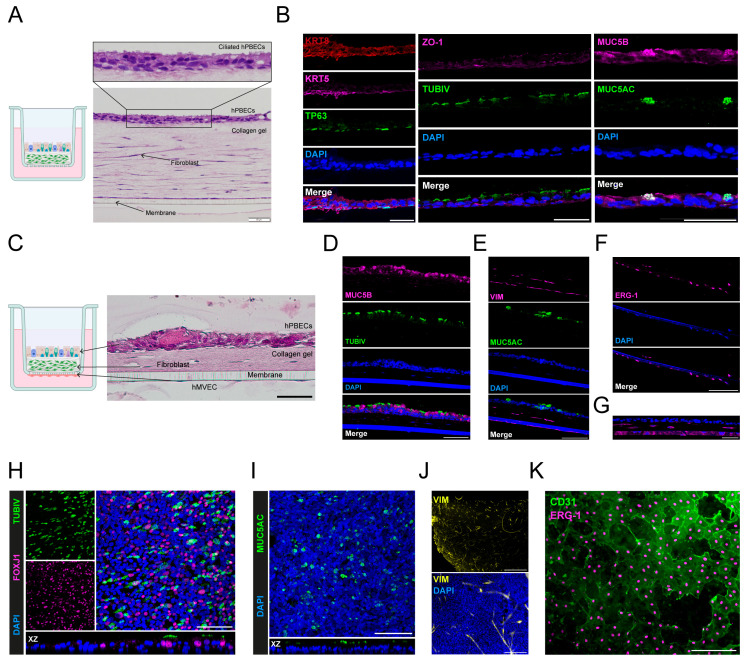
Development of a 3D human primary airway tri-culture model. (**A**) HE staining of a section of hPBECs and hLFs co-culture on an insert membrane after 14 days at ALI. Scale bar = 50 µm. On the left is a schematic illustration of the co-culture on a transwell insert: hLFs (green) were embedded in a collagen gel, and hPBECs were plated on top. (**B**) Staining of hPBECs grown on top of collagen gels in co-culture with hLFs. HPBECs expressed basal (TP63, KRT5, and KRT8), ciliated (TUBIV), and goblet cell markers (MUC5AC and MUC5B), and tight junctions (ZO-1) after 14 days at ALI. Scale bar = 50 µm. (**C**) HE staining of three-dimensional airway tri-culture after 14 days at ALI involving hPBECs, hLFs embedded in a collagen gel, and hMVECs plated on the basolateral side of the membrane (schematic illustration). Scale bar = 50 µm. (**D**–**G**) Tri-cultures after 14 days at ALI. (**D**) Confocal images of hPBECs stained with differentiation markers TUBIV (cilia) and MUC5B (goblet cells). Scale bar = 200 µm. (**E**) Confocal images showing goblet cell marker MUC5AC, and fibroblast and endothelial cell marker Vimentin (VIM). Scale bar = 100 µm. (**F**) Confocal images of a staining of the endothelial nuclear marker ERG-1. Scale bar = 200 µm. (**G**) Staining of VIM (magenta) and DAPI (blue) showing hLFs in the collagen gel and hMVECs on the membrane. Scale bar = 50 µm. H–K) Whole mount staining of tri-culture. (**H**) hPBECs ciliated (TUBIV and FOXJ1) and (**I**) goblet cells (MUC5AC). Scale bar = 100 µm. (**J**) hLFs grown in collagen gels stained with VIM. Top scale bar = 1000 µm, bottom scale bar = 200 µm. (**K**) Endothelial cells are positive for CD31 (green) and ERG-1 (magenta). Scale bar = 200 µm. For all experiments, N = 3–5. All schematic illustrations were made using BioRender.com.

**Table 1 ijms-26-05027-t001:** The scoring system summarizes the results of different media on the various cell types.

Cells/Medium	Complete KSFM	Complete BEGM	ECs Medium	DMEM + 10% FBS	Co-Culture Medium
hPBECs proliferation	++	+	−	−	−
hPBECs differentiation	+	++	−	−	+
ECs	−	−	++	+	+
hLFs	−	−	++	++	+
Tri-culture	−	−	−	−	+

**Legend:** ++ very good (high confluency, minimal cell patches, proper cell morphology), + good (medium confluency, some cell patches, decent cell morphology), and − bad (low confluency, many cell patches, bad cell morphology). For EC medium, we used EGM-2MV medium (for hMVECs) or ECFC-EGM medium (for hECFC cells).

**Table 2 ijms-26-05027-t002:** Summary of main cell culture media and coatings used for each primary lung cell type.

Primary Cell Type	Source Cells	Medium	Coating
hPBECs	Erasmus MC	Complete KSFM (growth medium)	hPBECs coating
		Complete BEGM (differentiation medium)	
hMVEC	Erasmus MC	EGM-2MV	EC coating
hMVEC-L	Lonza	EGM-2MV	No coating or EC coating
HUVEC	Lonza	HUVEC-EGM	No coating or EC coating
hECFCs	Erasmus MC	ECFC-EGM	EC coating
hLFs	Erasmus MC	DMEM + 10% FBS	No coating

## Data Availability

The data presented in this study are available at the request of the corresponding author for ethical reasons.

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
