# Peer review of "The Development of 3D Primary Co-Culture Models of the Human Airway"

_ijms, 2025, doi:10.3390/ijms26115027_

Round 1
Reviewer 1 Report
Comments and Suggestions for Authors
The authors of the paper entitled „The development of 3D primary co-culture models of the human airway“ developed a primary three-dimensional human airway model with higher complexity than the conventional models by using human, primary, bronchial epithelial cells, microvascular endothelial cells or endothelial colony forming cells, and lung fibroblasts. The authors tested different culture media and vessel coatings as well as cell culture hydrogels to optimize the co- and triple-culture conditions.
The authors performed a very comprehensive work on testing various media and their combinations. First, they tested whether the different media support cell growth and then they used the cell growth promoting medium and combinations in the differentiation of co- and triple 3D airway models. This approach is difficult to understand, because cell growth and differentiation are different processes that need different cell culture conditions. Why did the authors not use PneumaCult ALI as differentiation medium? This would be one of the differentiation media most often used.
Comment to Figure 2A: One can nicely see that the hPBECs grow better in PneumaCult Ex than KSFM. The morphology is much better.
The overall aim of the authors seems to be the development and use of 3D airway models to study respiratory diseases of children. The introduction at least creates this impression. However, the complex 3D models, they developed, are derived from adult tissues. The expectation raised by the introduction is therefore not fulfilled to a certain extent.
The authors cite several times “manuscript submitted” or “in preparation”. These references do not allow to obtain the specific information and should be replaced accordingly. See line 277, 484/5, 538
Line 70/71: „These models resemble the human airways, but also lack the complexity and physiology of the in vivo lung.“ What do the authors mean? Either the models resemble the human airways or they do not. Please, specify what is meant with lack of complexity and physiology?
Line 89/90: „Endothelial and mesenchymal cells, however, thrive in serum-rich medium (Table 1, Tables S2 and S3)“
The authors should consider that there are for both cell types defined serum-free media available that support the growth of the cells.
It would be better to show the table 1 when it is cited in the manuscript and not on page 8.
Several information are missing or should be specified:
Please include the supplier for PneumaCult Ex and PneumaCultALI.
What is the ECs medium see table 1?
What exactly is the hPBECs coating and what is co-culture coating?
Line 487: pre-coated 6-well plates. What is meant with pre-coated?
Line 508/9: Which cell culture coating was used? Please, specify.
Line 94: „Equal numbers of hPBECs were seeded, and cell density was monitored.“
The authors do not explain, how they assessed the cell density. An objective quantification of the cell density or better for the proliferation of the cells is needed.
The authors should provide the number of repetitions of each experiment?
Why are the age and sex not available for some of the donors? The missing information should be added, if possible.
Line 148: „thicker epithelium layers“ Although the epithelium seems thicker, it looks as there were not more cells compared to the other media.
The staining against zonula occludens-1 (ZO-1), which is located at the apical-lateral plasma membrane, shown in Fig. 3A, 4B, 5B, and 6B does not show the expected pattern. The authors should provide images with a better quality showing the specific ZO-1 staining.
Line 146-148: „Interestingly, we observed more ciliated (TUBIV: Figure 3A, Figure 4B, Figure S2A) and goblet (mucus secretion Figure 2B and MUC5AC: Figure 3B, Figure 4B) cells, and thicker epithelium layers (XZ images: Figure 3A-B) in co-culture medium than standard medium,“. According to the presented images, the amount of ciliated and goblet cells seems higher only in BEGM:EGM-2MV but not in BEGM:ECFC-EGM.
Line 21/22 and line 468: The authors write about “an underlying mucosa with fibroblasts”. However, mucosa is the outer layer and the fibroblasts are located within the subepithelial layer.
In my opinion, there are too many references included. The amount corresponds to a review article. In the methods and results part are even 35 references. The authors should recheck the references and reduce the number to the really necessary ones.
The following references are incomplete and should be corrected, in case they are included in the manuscript:
39. Gkatzis K, Taghizadeh S, Huh D, Stainier DYR, Bellusci S. Use of three-dimensional organoids and lung-on-a-chip methods to 727 study lung development, regeneration and disease. European Respiratory Journal2018.
60. You Y, Brody SL. Culture and Differentiation of Mouse Tracheal Epithelial Cells. 2012. p. 123-43.
Line 489: “according to Alphonse et al. protocol [63] 64” The citation should be corrected.
The webpage www.federa.org does not exist, please correct this information.
Reviewer 2 Report
Comments and Suggestions for Authors
The manuscript describes the steps necessary to develop a 3D co-culture model of the human airways consisting of epithelial, endothelial, and mesenchymal cells. The authors examine the optimal media composition of the tri-culture that enables the different cell types to grow and differentiate. Next, the study describes the optimal surface coating, followed by selecting the hydrogel for 3D culture to mimic the physiological structure of the human airways. Establishing an appropriate research model is well-motivated and offers a valuable platform for studying respiratory diseases and treatments. The manuscript is generally well-written, the methods are described clearly and sufficiently detailed, and validated by immunostaining. I only have minor comments.
The TEER values in Fig. 3E are too low to compare the different cells/conditions (line 187). This TEER range is usually within the measurement error range, since only higher TEER values are stable. Have you used biological/technical replicates for each reported TEER value?
Table 1 is in the wrong position and should be moved to the section before 2.2.
Unfortunately, the manuscript refers to unpublished sources twice (manuscript in preparation, lines 484 and 538). As stated, the hPBECs were isolated from resection tissue (line 485), but in line 500, it is noted that the hPBECs are thawed. Please describe the isolation and freezing protocol at least briefly.
The manuscript would benefit from discussing limitations that need to be addressed to improve models of the human airways further or acknowledging differences from in vivo tissue (e.g., absence of immune cells).
Unfortunately, not all information about the donors was accessible. Concluding the impact of donor specifics cannot be done.
In Table S1, the authors display whether or not the different cell types were obtained from each donor. What was the reason for not receiving all cell types from each donor? Experimental failure, not attempted, others…? For instance, the isolation of hPBECs was successful in 16/19 donors, speaking for a reproducible protocol, but for hMVECs only in 6/19 donors, which could suggest that the isolation protocol for hMVECs should be further improved. Please clarify.
Please include the abbreviation of WM used in Table S5.
Please include the abbreviation of IF used in Table S6.
Unfortunately, much of the necessary data is placed into the supplements. For instance, finding the optimal hydrogel mainly refers to the supplemental figures, impairing the readability as the reader has to switch between both.
In the discussion, the authors state that the model will gain better insights into the pathophysiology of pediatric respiratory diseases. Yet, cells were mainly derived from donors aged 45-73. Whether or not the results obtained with this model of likely senescent cells will also apply to pediatric pathologies has to be determined.
To highlight the study's novelty, the authors should clearly state what makes this 3D model stand out from currently used 3D models of human airways.
The sections „Data Availability Statement“,“ Acknowledgements“, and „Conflict of Interest“ are missing at the end.
Fig. S5: Typo „Complentary“
Round 2
Reviewer 1 Report
Comments and Suggestions for Authors
Thank you for revising the manuscript. Please make sure to upload images with high resolution demonstrating the ZO-1 staining clearly.
See comment: The staining against zonula occludens-1 (ZO-1), which is located at the apical-lateral plasma membrane, shown in Fig. 3A, 4B, 5B, and 6B does not show the expected pattern. The authors should provide images with a better quality showing the specific ZO-1 staining.
Authors' comment: We apologize for the loss of resolution in the PDF file, and will upload higher resolution images which are much more convincing.